# The Talented LncRNAs: Meshing into Transcriptional Regulatory Networks in Cancer

**DOI:** 10.3390/cancers15133433

**Published:** 2023-06-30

**Authors:** Dana Segal, Josée Dostie

**Affiliations:** 1Department of Biochemistry, McGill University, Montréal, QC H3G 1Y6, Canada; dana.segal@mail.mcgill.ca; 2Rosalind and Morris Goodman Cancer Institute, McGill University, Montréal, QC H3A 1A3, Canada

**Keywords:** cancer, long non-coding RNA, transcription factor, epigenetic, transcription, chromatin

## Abstract

**Simple Summary:**

Long non-coding RNAs (lncRNAs) are a large class of RNA molecules known to regulate the expression of genes. Although expressed in a highly tissue-specific manner, they appear to modulate virtually all pathways in the cell, either in normal or disease states. LncRNAs are often aberrantly expressed in cancers and can act as oncogenes or tumor suppressors. Here, we describe various mechanisms by which lncRNAs influence gene expression as they bind other RNAs, genomic DNA, or proteins. Particularly, we discuss their relationship with transcription factors as they act on pathways relevant to cancer. While much remains to be discovered about how they regulate genes, lncRNAs hold great therapeutic potential, especially in their transcription factor interactions.

**Abstract:**

As a group of diseases characterized by uncontrollable cell growth, cancer is highly multifaceted in how it overrides checkpoints controlling proliferation. Amongst the regulators of these checkpoints, long non-coding RNAs (lncRNAs) can have key roles in why natural biological processes go haywire. LncRNAs represent a large class of regulatory transcripts that can localize anywhere in cells. They were found to affect gene expression on many levels from transcription to mRNA translation and even protein stability. LncRNA participation in such control mechanisms can depend on cell context, with given transcripts sometimes acting as oncogenes or tumor suppressors. Importantly, the tissue-specificity and low expression levels of lncRNAs make them attractive therapeutic targets or biomarkers. Here, we review the various cellular processes affected by lncRNAs and outline molecular strategies they use to control gene expression, particularly in cancer and in relation to transcription factors.

## 1. Introduction

Genome-wide transcriptomic analyses revealed that over three quarters of the human genome can be transcribed and that approximately fifty percent of it may be read at a given time in a single cell [1]. RNAs are broadly classified as protein-coding (messenger RNAs; mRNAs) or non-coding (ncRNAs), the latter being the more diverse class that includes transcripts produced from annotated genes and between them. Gene-encoded ncRNAs can be further categorized into subgroups according to their cellular function, genomic location, and size [2]. Long non-coding RNAs (lncRNAs) are one such subtype longer than 200 nucleotides, encompassing a very large and diverse group of regulatory transcripts. With a few known exceptions [3], lncRNAs are transcribed by RNA polymerase II (POL II) and are thus 5′-capped, 3′-polyadenylated, and often spliced. Different groups estimate that lncRNAs could arise from at least as many (~30,000) or up to almost ten times more genes than those encoding proteins [4,5,6,7]. They are predicted to not code for protein from sequence features such as poor start codon context, uncharacteristic codon usage, and short length of detectable open reading frames that would otherwise trigger nonsense-mediated decay [8]. Curiously, polysome profiling has detected many lncRNAs [9,10] indicating that some do encode peptides [11,12]. Although most of these peptides are thought to lack function [10], some lncRNAs will thus likely require re-classification.

What makes lncRNA genes difficult to catalogue comes partly from their high cell type specificity and rather low expression levels compared to mRNAs, both of which may be linked to the sequence and structure of their genes [13]. One contributing feature is fewer overlapping transcription factor binding motifs at their core promoters, leading to greater specificity and lower expression levels [14]. Although lncRNA promoters are relatively well conserved [15], their gene bodies are much less subject to evolutionary constraints. This not only depreciates identification through sequence conservation but also impacts features expected in POL II genes, including weaker splicing signals that lead to poor splicing and less alternative splicing [15,16,17,18]. LncRNA genes tend to be shorter than those encoding mRNAs, have a lower exon count, and overall have longer introns and exons [19,20]. Interestingly, they are also enriched in transposable element (TE) sequences, which appear to have tremendously influenced lncRNA gene makeup throughout evolution [13,15,21]. In fact, TEs themselves were recently found to have lncRNA-like regulatory functions (more on this topic to follow). Another interesting finding about lncRNA genes is that they are often adjoining genes encoding transcription factors (TFs), suggesting intimate relationships between their function and regulation [22]. Below, we outline the various means through which lncRNAs control gene expression and further detail how the link between transcription factors and lncRNAs takes part in cancer pathways.

## 2. How LncRNAs Can Regulate Gene Expression in Cancer

### 2.1. Connecting LncRNAs to Cancer

Cancer refers to a group of diseases where cells evade regulatory checkpoints controlling growth, proliferation, and tissue invasion as natural processes go haywire, including controlled lncRNA expression. MALAT1, H19, and PCA3 are amongst the first lncRNAs found overexpressed in cancer, and along with many other transcripts, they were later confirmed to have important oncogenic roles (Table 1; [23,24,25]). HOTAIR is another historically significant lncRNA, as it is the first shown to epigenetically drive disease progression [26]. HOTAIR overexpression is in fact sufficient to increase cancer invasiveness and metastasis [27], and it was found to contribute to chemotherapy resistance in breast cancer [28].

Tumorigenic roles have also been found for lncRNAs originally implicated in fundamental cellular processes: the overexpression of XIST, famously known for its role in facilitating X chromosome inactivation, contributes to the pathology of different cancers [29] through various mechanisms involving protein binding partners [68,69,70,71]. Many lncRNAs were then linked to pathways often mutated in cancer, including the PI3K/AKT, p53, NF-κB, and Notch signaling pathways [72]. Interestingly, the computational prediction of lncRNAs that dysregulate cancer pathways indicates that most are likely disruptive only in specific tumor contexts. However, a few of them—including XIST—were found to synergistically influence cancer pathways in multiple tumor contexts together with other lncRNAs [73]. The context specificity of lncRNA function appears inherently linked to the polyvalence of their composition and whether they can engage in multiple mechanisms of action. In the sections below, we outline how lncRNAs can regulate gene expression and provide examples demonstrating such roles in cancer. These are summarized in Figure 1.

### 2.2. Direct Transcriptional Control

LncRNAs use various mechanisms to affect transcription, such as preventing the recruitment of POL II or TFs. Repression of the imprinted *Igf2r* gene by its antisense *Airn* lncRNA is an early example for this type of control [74] where *Airn* transcription on the paternal allele displaces POL II from the *Igf2r* promoter to silence the gene [75]. The GAS5 tumor-suppressor lncRNA also directly controls transcription. Its expression was found dysregulated in several cancers, and its tumor-suppressor activity is thought to arise from its ability to promote growth arrest and apoptosis and even inhibit cell migration in some cases [30]. GAS5 appears to exert its activity through many mechanisms: for instance, it was found to directly interact with the DNA binding domain of the glucocorticoid receptor (GR), preventing it from binding to and regulating transcription at the glucocorticoid response elements (GREs) of target genes [31]. GR is a constitutively and ubiquitously expressed TF that can either recruit or displace associated proteins when binding the GREs [76]. As such, context-specific GR chromatin binding leads to deliberate, cell type-specific gene expression changes, and how GAS5 might play into GR decoy activity has not yet been explored with respect to cancer-related gene expression.

Another interesting example of direct transcriptional control was found with the SINE-encoded B2 RNA in mouse. Although its shorter size (180 nt) and transcription by RNA polymerase III (RNA POL III) does not make it a lncRNA by conventional definition, it employs a mechanism of action analogous to that of lncRNAs. The B2 RNA was found to play an important role in the transient, general repression of POL II activity that occurs immediately upon heat shock and before heat shock-specific genes are induced [77,78]. B2 RNA transcription was rapidly induced by POL III concurrently to POL II repression, and accordingly, B2 RNA was found to bind the POL II σ-holoenzyme to prevent the formation of a functional pre-initiation complex, leading to the inhibition of POL II transcription initiation.

### 2.3. Epigenetic Control of Transcription

LncRNAs can also influence the likelihood of a gene’s transcription through epigenetic mechanisms that are not via direct interactions with POL II or TFs. The examples presented below are grouped based on their general influence on the chromatin landscape or three-dimensional (3D) architecture.

#### 2.3.1. By Associating with Chromatin-Modifying Complexes

Since the discovery that HOTAIR regulates gene transcription by recruiting histone-modifying complexes [79], a plethora of lncRNAs have been documented to recruit chromatin modifiers. Thus far, polycomb-group proteins (PcGs) have been the most commonly found chromatin modifiers associated with lncRNAs, and although this initially raised concern about interaction specificity [80], it ultimately highlighted the promiscuity of some protein–lncRNA interactions.

LncRNA–PcG complexes can reach their target genes via different means [81], including direct physical interaction of the RNA molecules with genomic DNA. Direct recruitment can occur through the formation of R-loops, which are those three-stranded nucleic acid structures wherein part of an RNA molecule is hybridized to a DNA strand, displacing the non-template strand. Given their inherent complementarity, lncRNA R-loops can form locally from their own genes to regulate overlapping genes, while non-paired RNA regions bind and recruit chromatin-modifying enzymes. Regulation of the *GATA3* locus by *GATA3-AS1* involves such a mechanism as the GATA3-AS1 lncRNA transcribed antisense to the *GATA3* gene forms an R-loop that tethers an MLL methyltransferase to activate transcription [82]. The lncRNA TARID is another example wherein transcription of the antisense *TARID* gene generates an R-loop at the *TCF21* promoter. The tethered TARID then binds GADD45A to recruit the DNA demethylating factor TET1, ultimately leading to *TCF21* activation and tumor-suppressor activity [32,33].

Another way in which lncRNAs can directly interact with genomic DNA to recruit chromatin-modifying complexes is through the formation of DNA–RNA triplexes (triplex-DNA): Rather than base-pairing with one genomic DNA strand, the RNA nestles into the groove of the DNA helix through Hoogsteen base-pairing [83]. The proto-oncogene khps1 lncRNA transcribed antisense to the *SPHK1* protein-coding gene is known to form triplex-DNA with that gene’s promoter and activates its expression by recruiting histone acetyltransferases CBP/p300 [34]. Epigenetic control by lncRNA/triplex-DNA does not always occur at gene promoters, nor is it restricted to cis regulation. For instance, the MEG3 lncRNA inhibits the transcription of *TGFBR1* in trans by facilitating H3K27me3 deposition and silencing through the formation of triplex-DNA and recruitment of the polycomb repressive complex 2 (PRC2) at one of its distal regulatory regions found to loop with its promoter [35].

LncRNAs can indeed act over long distances, with their influence found to spread beyond one gene, as HOTTIP, expressed from the *HOXA* cluster 5′ end, was found to upregulate the transcription of several surrounding genes by associating with the activating WDR5/MLL complex [84]. Further to serving as guides and scaffolds, lncRNAs can also globally influence chromatin modifiers by acting as decoys. YY1—a TF that can initiate, activate, or repress transcription depending on its binding context—is involved in transcription regulation genome-wide. Regulation by YY1 is influenced by linc-YY1, its antisense-transcribed lncRNA. Linc-YY1 interacts with the TF through its middle domain, which specifically evicts YY1-PRC2 complexes from target promoters, resulting in gene activation [85].

In addition to influencing the recruitment of chromatin-modifying proteins to genomic DNA, lncRNAs also determine the functional composition of RNPs as they can have more than one protein-binding domain and thus recruit varying combinations of distinct proteins. HOTAIR was the first reported lncRNA to bind more than one chromatin-binding complex; it was found to associate with PRC2 at its 5′ end and the LSD1/coREST/REST complex at its 3′ end, both of which contribute to transcription inhibition by coordinating H3K27 methylation with H3K4 demethylation, respectively [26]. Since then, the affinity of lncRNAs for chromatin-modifying complexes—especially the PRC2 complex—and other transcriptional regulators has been well documented in cancer. Lnc-LBCS was shown to inhibit the self-renewal and chemoresistance of bladder cancer stem cells through the epigenetic silencing of *SOX2,* which was partly through H3K27me3 deposition facilitated by binding the EZH2 component of PRC2 and hnRNPK to the *SOX2* promoter via triplex-DNA formation. While hnRNPK—an essential RNA/DNA-binding protein with critical roles in several cancers [86]—was previously found to directly mediate transcription [87], binding to lnc-LBCS was suggested to confer it a gene-repressive function [36], exemplifying the role lncRNAs can have in determining context-specificity.

#### 2.3.2. By Modulating Spatial Chromatin Organization

Key research developments have established an important role for lncRNAs in 3D chromatin organization as both influence transcription as well as various other genome activities including X-chromosome inactivation (XCI), imprinting, DNA damage repair, and DNA replication. First, genome organization is now studied both by microscopy and with genome-wide high-resolution mapping techniques, including Hi-C and ChIA-PET. These techniques were designed to capture chromatin organization from pair-wise DNA contacts—or with more recently developed methods that additionally measure multivalent contacts, such as GAM (genome architecture mapping) and SPRITE (split-pool recognition of interactions by tag extension) [88,89,90,91,92]. Importantly, recently developed technologies, HiChIRP and RD-SPRITE, directly measure RNA-associated chromatin conformation genome-wide [92,93,94]. Second, lncRNA–chromatin interactions have been found for hundreds of different transcripts across the genome [95,96,97,98,99,100]. Third, lncRNAs can regulate transcription through different epigenetic processes including DNA methylation, nucleosome positioning, chromatin remodeling, and 3D chromatin organization [101]. LncRNAs regulating spatial genome organization have been found at any resolution scale; at high-resolution, they are known to promote or prevent the formation of chromatin loops between control DNA elements such as enhancers, promoters, and insulators [102]. They can also participate in the spatial clustering of DNA elements [103], alter TAD structure [104], drive LLPS to form POLII-containing transcriptional condensates around which chromatin organizes [92,105], or act as chromatin organizing hubs [106].

At lower resolution, lncRNAs can act as nuclear organization factors on their own or as part of nuclear subcompartments. For example, NEAT1 (nuclear paraspeckle assembly transcript 1) is a lncRNA that localizes to paraspeckles, which are transcription-dependent, self-organizing, membrane-less nuclear organelles [107]. Paraspeckles are thought to serve as sponges that modulate the availability of active molecules to regulate distinct processes such as transcription [37], the nuclear retention of A to I edited RNAs [38,108], an active chromatin state and miRNA biogenesis [39]. NEAT1 is highly expressed across tissues and frequently overexpressed in human tumors, which correlates with worse survival rate in cancer patients [40]. Another highly and ubiquitously expressed nuclear lncRNA important for nuclear organization and gene expression is MALAT1 (metastasis-associated lung adenocarcinoma transcript 1) [41], which is found in nuclear speckles—large nuclear structures also known as interchromatin granule clusters (IGCs), enriched in TFs and splicing factors (SFs) that coordinate transcriptional and post-transcriptional gene regulation [42]. MALAT1 was shown to relocate genes from repressed polycomb bodies to transcriptionally active regions, presumably by interacting with PcG proteins to relieve repression [43,44]. Through various mechanisms, dysregulated MALAT1 expression affects all kinds of human cancers as an oncogene or tumor suppressor [45], acting upon cellular processes such as alternative splicing [46], epithelial to mesenchymal transition (EMT) [47], apoptosis [48], and autophagy [49].

Xist (X-inactive specific transcript) is yet another important nuclear lncRNA that controls gene expression partly through changes in 3D chromatin organization [109]. Xist is indispensable for XCI (X Chromosome Inactivation)—the silencing of one X chromosome in females that compensates for gene dosage between sexes [110,111]. Xist expression from one X chromosome triggers the inactivation of most of its ~1000 genes by recruiting different factors, including PcG complexes, to trigger the formation of a facultatively heterochromatic chromosome—or “Barr body” [68,112,113,114]. Xi adopts an unusual conformation as two large compacted domains are kept separate by a ~200 kb linker region [115], which is distinct from the usual organization of autosomes and the active X (Xa) into hierarchically organized TADs [116,117]. While the mechanism behind Xi folding is not yet fully understood, several findings support a model in which PcG proteins molecularly prime sites to direct 3D chromosome organization, as has previously been reported in autosomes [118,119]; they are already bound at X-linked regions near the Xic prior to Xist spreading [120], and those regions are first targeted at the onset of XCI [121]. An interesting notion is that Xist together with PcG complexes might form membrane-less bodies that drive Xi 3D reorganization through LLPS, which is bolstered by studies reporting that Xist associates with proteins prone to mediate phase separation [122,123], and that LLPS was suggested to partly underlie heterochromatin formation [124]. Finally, similarly to NEAT1 and MALAT1, aberrant Xist expression has been reported in numerous cancer types where it is mainly found overexpressed and acting as an oncogene by interfering with miRNA regulation [29].

As illustrated by the examples above, lncRNAs can influence 3D chromatin organization in different ways. For instance, the Firre lncRNA encoded on the X chromosome escapes XCI to act as an important genome organizer, reportedly anchoring the Xi to the nucleolus [125], helping maintain Xi silencing [126], and physically linking the *Firre* genomic locus to several regions on other chromosomes [127]. Conversely, a large-scale study suggests that lncRNA–chromatin binding sites are themselves determined by chromosome conformation, in that lncRNAs may act as scaffolds to direct regulatory factors to promoters and enhancers genome-wide [128]. At high resolution, they are known to promote or prevent the formation of chromatin loops between control DNA elements such as enhancers, promoters, and insulators [102]. They can also participate in the spatial clustering of DNA elements [103], alter the TAD structure [104], drive LLPS to form POLII-containing transcriptional condensates around which chromatin organizes [92,105,129], or act as chromatin-organizing hubs [106].

The mechanisms by which lncRNAs alter chromatin conformation is either through direct DNA recruitment or through protein association. Changes in chromatin organization have been linked to R-loop formation [104] and to triplex-forming RNA that can guide 3D genome organization [130,131]. LncRNAs were also found to interact with chromatin architectural proteins. For instance, Firre appears to promote interchromosomal physical proximity by interacting with hnRNPU, which is a nuclear matrix-interacting protein also known for its affinity to “scaffold/matrix attachment regions” (S/MARs) [127]. Importantly, CTCF (CCCTC-binding factor) [132], which acts as an insulator and master regulator of genome organization [133], was shown to bind RNA without specificity [134,135]. CTCF’s regulation of lncRNAs pertains to cancer, as it interacts with Wrap53, a natural antisense transcript to p53 that regulates its expression upon DNA damage and in cancer cells [136,137]. Although merely suggested in original studies, the binding of CTCF to RNA was later revealed essential for its role in 3D genome organization as it is required for its ability to form chromatin loops, insulate domains, and mediate long-range contacts [134,135,138,139]. Taken together, these studies illustrate how lncRNAs use a variety of mechanisms to modulate 3D chromatin organization throughout its hierarchical levels as a means to influence transcription.

### 2.4. Control of Splicing

After transcription, lncRNAs continue to control gene expression by influencing alternative splicing (AS) [140]. In humans, over 90% of intron-containing genes undergo AS, which confers most of a cell’s proteomic diversity [141]. As such, aberrant AS is a hallmark of cancer, and evidence suggests that lncRNAs are important AS regulators in healthy and diseased cells [142]. LncRNAs impact AS through both direct and indirect mechanisms; for example, those that change chromatin context affecting POLII elongation rates will indirectly influence AS [143,144]. LncRNAs use various strategies to directly control the splicing of specific genes, which we outline and provide examples for below.

#### 2.4.1. By Modifying Chromatin

Epigenetic chromatin modification is known to influence AS, and cases have arisen where lncRNA-directed epigenetic changes play a part [140]. The cell type-specific AS of exon IIIb in *FGFR2* is a good example where in mesenchymal cells, exon IIIb is excluded because the negative splicing regulator PTB is recruited by the H3K36me2/3-binding protein MRG15. However in epithelial cells, this exon is retained due to the production of a nuclear antisense lncRNA from within *FGFR2* that recruits PcG protein and the H3K36me2/3 demethylase KDM2a, which prevents binding of the MRG15-PTB complex [145].

LncRNAs can also affect AS by altering 3D chromatin organization. One way by which such a mechanism has been used is by altering CTCF chromatin association. Although CTCF binding on its own can affect AS by inducing POL II pausing and exon inclusion [146], whether or not CTCF can bind may also alter its ability to engage in chromatin looping and domain formation [147]. One such example can be found at the *Protocadherin a* gene cluster, where first exon selection is dependent on the expression of its antisense lncRNA, which by promoting local DNA demethylation allows CTCF binding and long-range looping with a distal enhancer that will drive transcription from the selected site [148]. Given the great importance of AS defects and frequent lncRNA dysregulation in cancer [149,150], it will be interesting to assess how they intersect in disease onset and progression.

#### 2.4.2. By Forming RNA–DNA Interactions

To date, a subclass of lncRNAs—the circular RNAs (circRNAs)—was reported to directly regulate AS by interacting with genomic DNA. CircRNAs are produced by a non-canonical AS back-splicing process that occurs mostly at the middle exons of pre-mRNAs [151]. Like other lncRNAs, circRNAs are highly cell-type specific and regulate gene expression at various stages from transcription to signaling and translation, depending on their cellular localization. As such, they are important regulators of normal physiological processes, with key roles in development and disease [152,153]. One case of direct AS control by circRNAs was reported at the *SEPALLATA3* gene in *Arabidopsis thaliana,* where a circRNA derived from its exon 6 forms an R-loop with the corresponding DNA sequence, causing transcriptional pausing, specific SF recruitment, and exon 6 skipping in a shorter isoform responsible for homeotic phenotypes [154]. While circRNA transcripts have been found to act via different mechanisms in various cancer contexts, their back-splicing ability has yet to be implicated as such [153].

#### 2.4.3. By Influencing Splicing Factor Activity

LncRNAs also can have a broader impact on AS by modulating the activity and/or localization of SFs. One such example is the regulation of the SR family SFs phosphorylation and localization by MALAT1 in nuclear speckles [46]. MALAT1 was also reported to increase the abundance of SR SFs: such is the case in hepatocellular carcinoma, where it acts as a proto-oncogene by increasing SRSF1 expression, leading to high levels of anti-apoptotic S6K1 isoforms [50]. In colorectal cancer, the binding of MALAT1 to SFPQ releases the PTBP2 oncogene from an SFPQ/PTBP2 complex, which promotes tumor growth and metastasis [51]. Nuclear paraspeckles can also retain RNAs or proteins to control transcription and splicing: NEAT1—the core lncRNA component of paraspeckles—can differentially regulate AS based on cell context by forming RNP complexes with both core and cell-type specific proteins [155]. For instance, NEAT1 can generally control alternative splicing by sequestering SFPQ within paraspeckles, which generally restricts pro-apoptotic events while specifically allowing IL-8 transcription upon immune stimulation [52,53]. NEAT1 is also known to tether SRp40 into paraspeckles to alter the abundance of the PPARg isoforms as a major driver of adipocyte differentiation [54]. Beyond nuclear bodies, another lncRNA named PNCTR has been linked to general splicing activity, as it can bind and sequester the hnRNP-family PTBP1 to prevent its participation in pro-apoptotic splicing events in cancers [55].

#### 2.4.4. By Forming RNA–RNA Hybrids

LncRNAs can modulate splicing using different mechanisms involving direct contacts with RNAs. A lncRNA can directly bind a pre-mRNA, as does the SAF lncRNA shown to bind its antisense pre-mRNA encoding FAS as well as splicing factor 45 (SPF45) to induce exon skipping. The resulting FAS mRNA variant lacks exon 6 and translates into soluble Fas that retains its ability to bind the Fas ligand and blocks interaction with the Fas-receptor and pro-apoptotic signaling in human cancer cells [56]. RNA–RNA interaction is employed through a different mechanism for ZEB2, which encodes a protein that represses E-cadherin and prevents Snail-induced EMT. Binding of the ZEB2 natural antisense transcript (NAT) to the ZEB2 pre-mRNA covers a 5′ splice site in its 5′ UTR, leading to the inclusion of an IRES-containing intron from which translation can be initiated [57]. Using yet another mechanism, the BC200 lncRNA, bearing a 17-nucleotide complementary sequence to the BCL-x pre-mRNA, can recruit hnRNPA2/B1 to prevent association with Sam68, which favors the expression of BCL-xL over the short BCL-xS variant. As BCL-xS favors apoptotic conditions, BC200 is often upregulated through estrogen signaling in breast cancer to have an oncogenic effect [156].

RNA contacts are not restricted to the specific splicing regulation of certain mRNAs, as exemplified by MALAT1′s partial hybridization to many nascent pre-mRNAs to participate in splicing through SF recruitment [157]. Interestingly, MALAT1 was also found to interact with U1snRNA [157], suggesting that it might not only directly influence splicing but may possibly also affect mRNA 3′ end formation by regulating telescripting, the cotranscriptional suppression of premature mRNA 3′ end cleavage and polyadenylation [158].

### 2.5. Translational Control

#### 2.5.1. By miRNA Sponging or as Competing Endogenous RNAs (ceRNAs)

A most frequently reported activity for cytoplasmic lncRNAs is their action as competing endogenous RNAs (ceRNAs) to effectively “sponge” microRNAs away from target mRNAs. As miRNAs can induce translation repression and/or promote mRNA decay, miRNA sponging by lncRNAs generally makes them positive mRNA regulators that lead to higher protein levels [159]. Initially when miRNAs were observed to inhibit mRNA translation, they were thought to function by preventing 80S complex formation [160] or by blocking elongation [161]; however, the initiation phase was quickly identified as the main step targeted by various mechanisms [159]. The immense and continuously increasing reported examples of miRNA sponging points to it as a widespread mechanism to control many—if not all—cellular pathways in normal or disease conditions. In fact, lncRNA/miRNA/mRNA regulatory axes have already been implicated in many aspects of cancer development, including angiogenesis, neovascularization, and vasculogenic mimicry [162]. Among these innumerable examples, the TUG1 lncRNA sponges many miRNAs in multiple regulatory axes and cancers: the TUG1/miR-299/VEGF-A axis increases angiogenesis in glioblastoma [58], while the TUG1/mir-34a-5p/VEGF-A axis is thought to contribute to hypervascularity in hepatoblastoma [59]. Moreover, the TUG1 lncRNA was found to sponge miR-145, which is a reported tumor suppressor that helps regulate SOX2 and MYC mRNA levels and thus upregulates the abundance of these key stemness-associated TFs in glioma [60]. Another lncRNA—HOXA-AS2—was shown to sponge miR-373, leading to the upregulation of EGFR and greater density of vasculogenic mimicry [61]. LncRNA binding to miRNAs does not merely sponge them but can also degrade them to enhance mRNA translation: indeed, one of many examples for this type of control occurs when the OIP5-AS1 lncRNA binds miR-7 during human myogenesis, leading to miR-7 degradation through target-directed miRNA degradation (TDMD) and thereby higher MYMX target mRNA translation [163].

Most studies report derepressed translation as a result of lncRNA sponging of miRNAs, yet it can also potentially lead to translation inhibition as some miRNAs act as decoys to translation inhibition proteins. Such is the case for miR-328 shown to bind hnRNP E2 in a way that prevents it from binding to and repressing translation of the CEBPA mRNA in leukemic blasts [164]. While not yet demonstrated, a lncRNA’s sponging of such miRNA would elicit increased translational output.

#### 2.5.2. By Controlling the Subcellular Localization of mRNAs

LncRNAs can bind other RNAs beyond miRNAs, such as mRNAs which influence their subcellular localization and translational output [165]. Several examples of this mode of translational control exist, such as between Uchl1-AS1 and Uchl1, when upon rapamycin treatment, the Uchl1-AS1 lncRNA translocates from the nucleus to the cytoplasm, where it binds the Uchl1 mRNA and promotes translation through ribosome recruitment [166]. In contrast, nuclear translocation and binding of the AdipoQ–AS lncRNA to the AdipoQ mRNA represses translation in the cytoplasm and thereby prevents adipogenesis [167]. The abundance of lncRNA–RNA interactions and their perception as subtle “fine-tuners” of gene regulation likely contribute to underappreciation in cancer research; however, the well-documented effects of this mechanism on cell proliferation and survival significantly bolsters their potential as biomarkers and therapeutic targets.

#### 2.5.3. By Regulating Translation Factors

Several reports describe lncRNAs directly binding translation factors to regulate mRNA translation. In most cases, lncRNAs target the rate-limiting step of translation—the initiation phase—where it often interferes with its limiting factor: eIF4E [168]. GAS5 was reported to decrease c-myc translation via eIF4E binding in lymphoma [62]. The SNHG1 and SNHG4 lncRNAs also bind eIF4E in mantle cell lymphoma [169]. Other lncRNAs appear to target additional components of the eIF4F complex beyond eIF4E; ribonucleoprotein complexes containing the lncRNA treRNA were shown to bind eIF4G1 and suppress E-cadherin translation [63], while the brain-specific BC1 lncRNA was reported to bind eIF4A and to negatively regulate translation of the polyA-binding protein PABP [170].

### 2.6. Post-Transcriptional/Translational Control

#### 2.6.1. By Regulating mRNA or Protein Stability

LncRNAs use various mechanisms to alter the stability of mRNAs and proteins as gene expression regulation. As previously mentioned, mRNAs under the influence of miRNA-mediated degradation may be controlled indirectly by ceRNAs. Direct mRNA cleavage by miRNAs seem to be more the exception than the rule, as it occurs for ~6% of mRNAs targeted by miRNAs, while the rest appear to become destabilized due to translation inhibition [159,171].

LncRNAs can base-pair with their mRNA targets to also recruit protein complexes that impact their stability. Staufen1 (STAU1) is one such protein, which is recruited through base-paired Alu sequences to bind resulting double-stranded lncRNA/mRNA hybrids, leading to STAU1-mediated mRNA decay [172]. While in principle, any lncRNA base-paired through Alu sequences can bind STAU1 to induce mRNA degradation, STAU1 has dual modalities and can conversely be recruited through other paired sequences to elicit mRNA stabilization. For instance, the TINCR lncRNA contains 25-nucleotide motifs that can base-pair and recruit STAU1 on many different mRNAs to stabilize them and maintain their expression during epidermal differentiation [173].

LncRNAs were also found to alter mRNA levels without directly binding them but by interacting with proteins that customarily control mRNA stability. An example is how CAAlnc1 acts as decoy for HuR; in binding HuR, CAAlnc1 prevents the degradation of hundreds of adipogenesis-related transcripts, some of which encode adipogenic TFs [174]. Another example is the sequestration of Pumilio proteins by the lncRNA NORAD [64]. Pumilio proteins bind mRNA 3′ UTRs in a sequence-specific manner to promote decay through deadenylation and decapping. NORAD expression, which is often dysregulated in cancer [65], is highly induced upon DNA damage to act as a decoy to Pumilio proteins, thereby stabilizing mRNAs involved in genomic stability [64].

While lncRNAs can bind proteins to alter mRNA stability, they can also affect the stability of proteins themselves. This can occur directly, as for lncRNA SNHG15, as it binds the Slug protein’s zinc finger domain to prevent its ubiquitination, thereby protecting it from proteasome-mediated degradation and promoting colon cancer cell proliferation [66]. Protein stabilization can also occur indirectly through post-translational modifications: the lncRNA PSTAR exhibits tumor-suppressor activity in hepatocellular carcinoma by interacting with hnRNPK and enhancing its SUMOylation; this strengthens its stabilizing interaction with p53 and ultimately leads to cell-cycle arrest [67].

#### 2.6.2. By Controlling the Subcellular Localization of Proteins

In addition to MALAT1, which as detailed above controls SR splicing factor distribution, some lncRNAs can regulate the cellular localization of specific proteins. For example, while the NRON lncRNA keeps NFAT in the cytoplasm in resting T-cells, it migrates to the nucleoplasm upon T-cell activation [165]. Another example is interaction of the GAS5 lncRNA with YAP, which is a transcription coregulator of the hippo signaling pathway. In colorectal cancer, the binding of GAS5 to the WW domain of YAP was reported to facilitate its translocation from the nucleus to the cytoplasm where its ubiquitination and degradation is promoted by phosphorylation [175].

## 3. LncRNA and Transcription Factor Crosstalk in Cancer

### 3.1. Transcription Factor Regulation by/of LncRNAs

As POL II-transcribed genes, lncRNAs are subject to much of the same transcriptional regulation as their protein-coding counterparts. Indeed, despite minor, but distinguishing, structural differences between lncRNA and mRNA genes [14,176], they appear to be controlled by—and enmeshed in—the same transcription factor networks (Table 2). A lncRNA both regulating and regulated by a TF is PANDA [177]. Transcribed from a single exon antisense to the *CDKN1A/p21* promoter region, PANDA is induced by p53 upon DNA damage and is often overexpressed in human cancers, granting its use as a diagnostic biomarker [178]. PANDA was shown to exert anti-apoptotic activity by binding and sequestering the NF-YA TF, preventing p53-mediated activation of pro-apoptotic p53 target genes [177]. PANDA was reported to promote cell proliferation by also binding Scaffold Attachment Factor A and recruiting PRCs to repress senescence-promoting genes [179]. As such, PANDA appears embedded in a transcriptional regulatory network through its association with TFs to impress upon cell cycle progression, senescence, and apoptosis.

Keeping with the ubiquity of feedback regulation in cellular biology, lncRNA promoters can also be enriched in binding sites for the TFs they in turn regulate. This is the case for the AK028326/MIAT/Gomafu lncRNA, which is both induced by and co-activates the Oct4 TF in embryonic stem cells [195]. This lncRNA has since been linked to many different cancer types and considered a potentially valuable biomarker [196], which makes sense considering its regulatory relationship with such a key stemness factor in cancer [197]. A more recent example of this kind of mutual regulation was found in basal-like breast cancer between KLF5 and KPRT4 [180]. KLF5—a key oncogenic TF in this cancer type—binds and activates the KPRT4 lncRNA gene promoter. In turn, the KPRT4 lncRNA forms a triplex with DNA at the 3′ end of the KLF5 gene, while it recruits YB-1 to its 5′ end [180].

LncRNA–TF crosstalk can go beyond transcript-mediated regulation and involve the lncRNA genes themselves; as exemplified by PVT1, lncRNA promoters and/or their gene bodies can regulate TF expression. *PVT1* is a lncRNA gene known to regulate the expression of *MYC*, which is located around 55 kb upstream on the same chromosome. *PVT1*’s normal expression is monoallelic, and as its promoter competes with the *MYC* promoter for intragenic *PVT1* enhancers on its single expressed allele, in doing so, it acts as a tumor suppressor by regulating the release of *MYC* transcriptional pausing [198]. However, as it resides in a genomic region often rearranged in cancers, the *PVT1* gene has been found to act as an oncogene under certain conditions, including its amplification in extrachromosomal DNA (ecDNA), and it drives *MYC* transcription in trans [199].

### 3.2. Molecular Function of LncRNAs in Transcription Regulation

Although lncRNAs and TFs have been independently implicated in cancer, increasing evidence suggests that they also do so together as RNPs. The identification of functional lncRNA–TF complexes frequently come from studies of aberrantly expressed lncRNAs, where gene targets were identified and RNA immunoprecipitation assays were used to validate physical interactions between the lncRNA and a TF also regulating those genes. Such studies usually proceed with the knockdown and/or overexpression experiments that demonstrate cooperativity between the lncRNA and TF for proper gene expression, roles in cancer pathways, and effects on cell proliferation, differentiation, and migration. LncRNA–TF RNPs can either positively or negatively regulate target gene transcription. For example, the HAND2-AS1 lncRNA was found to bind E2F4 at the *C16orf74* promoter to downregulate its expression and repress cervical cancer progression. In contrast, HNF1A-AS1 was found to bind PBX3 to increase *OTX1* expression and promote angiogenesis through ERK/MAPK signaling in colon cancer [181,182]. While many studies have provided general insight into lncRNA function, other studies delve further into molecular details of the mechanisms employed by lncRNA–TF complexes to regulate genes. In general terms, lncRNAs can participate in control mechanisms as decoys, guides, scaffolds, or modulators of chromatin folding and combinations thereof. These strategies are summarized below and illustrated in Figure 2.

#### 3.2.1. As Decoys

The binding of a lncRNA to a TF can prevent the protein from binding either another nucleic acid or protein. In addition to the aforementioned examples of linc-YY1/YY1 and GAS5/GR, where lncRNAs regulate transcription factor binding to chromatin, the MAGI2-AS3 lncRNA was found to exert tumor-suppressive and anti-angiogenic activities in clear cell renal cell carcinoma by interacting with and reducing the enrichment of the HEY1 TF at the *ACY1* promoter region [183]. Furthermore, the p53-activated TP53TG1 lncRNA was shown to bind and prevent the nuclear localization of YBX1, which usually activates oncogenes and many other genes, including PI3K. As AKT and MDM2 are known targets of PI3K phosphorylation, TP53TG1 expression thus protects p53 degradation and contributes to a p53 positive feedback loop [184].

#### 3.2.2. As Guides

LncRNAs can help TFs find genomic DNA target regions as well as prevent it. Whereas MAGI2-AS3 reduces the chromatin association of TF HEY1 [183], lncRNA-HIT does the opposite for its TF binding partner: this lncRNA was indeed shown to increase the binding and stability of the ZEB1 TF at the CDH1 promoter, leading to increased proliferation and migration in non-small cell lung cancer [185]. Mechanistically, lncRNAs can form R-loops with genomic DNA and recruit TFs to control gene transcription, epigenetic modifiers or other proteins, as reported for TARID [33]. A mechanism similar to TARID’s control of *TCF21* was reported for the lncRNA SATB2-AS1 at the *SATB2* gene, where it recruits WDR5 and GADD45A, eliciting gene activation that is thought to inhibit metastasis in colorectal cancer [186].

As previously discussed, lncRNAs can also directly bind genomic DNA as triplex-DNA, as does the LNMAT2 lncRNA at the *PROX1* TF promoter, where it also physically associates with hnRNPA2/B1 to promote *PROX1* transcriptional activation in bladder cancer [187]. Interestingly, triplex-DNA formation coupled to TF-binding by a lncRNA does not always lead to increased interaction of the TF to its target gene. The CISAL lncRNA, for example, is known to bind the *BRCA1* promoter and simultaneously bind the GABPA TF; however, the binding of GABPA to CISAL actually prevents the TF from binding the *BRCA1* promoter. As such, CISAL contributes to the repression of the tumor-suppressor gene and better disease outcome in tongue squamous cell carcinoma. This study proposes a novel model by which lncRNAs prevent TFs from binding target DNA that does not involve competitive binding or acting as decoys [188].

Alternatively to R-loop or triplex-DNA formation, RNA–RNA interaction may be a potentially important mechanism underlying lncRNA–TF recruitment to target genomic loci. Steitz and colleagues first demonstrated this with the ncRNA EBER2, which is a ~170 bp ncRNA expressed by the Epstein–Barr virus that interacts with the PAX5 TF. Using CHART, they found that PAX5 and the ncRNA EBER2 co-localize at PAX5 binding sites, and that RNA–RNA interaction with nascent transcripts guided such localization, which was later supported by demonstrating an in vitro interaction between ncRNA and nascent transcripts from such loci [200,201,202]. While not yet reported for lncRNAs, RNA–RNA interactions—perhaps with nascent transcripts—could represent an alternative mechanism for lncRNA-guided TF localization in cancer contexts.

#### 3.2.3. As Scaffolds

LncRNAs can act as architectural platforms to facilitate the interaction of multiple proteins and/or nucleic acids; they have been shown to do so with TFs, such as DINO, which supports p53 tetramerization and recruitment at p53 target genes [189]. Similarly to HOTAIR, the ANRIL lncRNA was reported to interact with PRC2 and PRC1 to regulate the expression of neighboring tumor suppressor gene *CDKN2A/B*, thereby affecting cell senescence, proliferation, aging, and apoptosis [203,204]. In prostate cancer, ANRIL specifically binds the PRC1 subunit CBX7 to silence the *INK4a* locus from which the lncRNA is expressed [190]. In trans, ANRIL can recruit PRC2 to *KLF2* and *p21* promoters to silence them in NSCLC [191]. ANRIL is also a TF-binding lncRNA, as its presence is required for YY1 binding at the *IL6* and *IL8* promoters in human endothelial cells [192]. This dual association with TFs and PcG complexes could be a yet unexplored co-operational mechanism through which ANRIL affects gene expression in cancer contexts. Indeed, the TUG1 lncRNA does just that, as it acts as a scaffold for PRC2 and YY1 to repress YY1-target genes associated with neuronal differentiation in glioma [60]. As lncRNAs can combine TF binding with genomic DNA or chromatin-modifying complex associations, it is possible that all three might occur within a single transcript to confer target specificity to epigenetic-modifying complexes.

#### 3.2.4. As Modulators of Chromatin Folding

As with general transcription, the act of lncRNA transcription itself likely plays an important role in modulating the 3D chromatin landscape to impart transcriptional regulation [176]. As also detailed above, lncRNAs can influence long-range chromatin contacts, some of which are important to the transcription of mRNAs encoding TFs integral to cancer pathology. In human colorectal cancer, the lncRNA CCAT1-L was implicated in chromatin looping at the *MYC* locus. CCAT1-L is transcribed from a super-enhancer region located 515 kb upstream of the *MYC* gene. It upregulates *MYC* by mediating long-range chromatin contacts with its promoter by binding to CTCF and either locally concentrating or allosterically modifying its association at the *MYC* locus [193]. CCAT1-L is not the only lncRNA involved in *MYC* enhancer–promoter interactions: the MYMLR lncRNA transcribed antisense to *MYC* also facilitates these contacts to upregulate the proto-oncogene in colorectal cancer. This interaction, between the *MYC* promoter and a super enhancer region located 550 kb upstream of *MYC,* relies upon MYMLR’s binding of the oncogenic PCBP2 splicing factor. Interestingly, it might be the same upstream enhancer region that is looping with the *MYC* promoter and implicated with both MYMLR and CCAT1-L activity [194]. As CCAT1-L and MYMLR can both be found aberrantly expressed to influence chromatin looping in certain cancer types, it could be that this mechanism might underly the context specificity and convergence of key regulatory pathways that modulate chromatin dynamics in cancer cells.

## 4. Cancer and the LncRNA-like Activity of Transposable Elements

TEs are mobile repetitive DNA sequences capable of moving from one genomic location to another. This migratory ability has enabled them to expand in our genome over the last 80 Myr of primate evolution, such that they now represent at least 45% of our DNA sequence [205]. TEs have substantially shaped human genome evolution in diverse ways, including promoting gene formation, exonization and the regulation of nearby genes [206,207]. One of the most provocative examples remains the recruitment of Rag1 and Rag2 to catalyze V(D)J somatic recombination in vertebrate immune systems, which is thought to originate from a transposition event over 500 Ma [208,209]. Most often, however, TEs have influenced the composition and regulation of transcripts. This is especially true for lncRNA genes, which have been populated by various types of TEs, most often retrotransposons [210]. In fact, several studies find much greater TE sequence enrichment in lncRNAs than mRNAs: approximately 40% of lncRNA sequences is estimated to come from TEs compared to 5.5% in mRNAs [21,211,212]. Moreover, TE insertion is thought to provide functional domains—regions that bind either proteins or other nucleic acids—to lncRNAs [213]. A good example of this so-called RIDL (Repeat Insertion Domains of LncRNAs) hypothesis is the discovery that a point mutation in a TE-derived domain of a lncRNA is associated with infantile encephalopathy [214]. Another example supporting the RIDL hypothesis is found in the aforementioned ANRIL lncRNA, which appears to have domesticated multiple transposons and transformed into exons that subsequently remained highly conserved in simians [215].

The ways TEs contribute to the emergence of lncRNA features, regulation, and regulatory functions seem to vary greatly, and to date, at least two TEs themselves were shown to have lncRNA-like activities: human long interspersed nuclear element 1 (LINE1) [216] and the B2 SINE RNA in mouse [217]. LINE1 is a type of non-LTR retrotransposon that make up approximately 18% of the human genome and the only known TE of its kind still capable of retro-transposition. Yet, only 80–100 of the estimated 870,000 human copies appear active, while most lie dormant as they lack necessary sequences to jump [210,218]. LINE1 is expressed at high levels in germ cells and embryos [219], where it was found to repress the *Dux* TF gene and induce rRNA expression by interacting with a Nucleolin-KAP1 protein complex [216]. This lncRNA-like activity is even more interesting given its greater mobilization and expression in many different cancer types [220], with *LINE1* hypomethylation in fact sometimes used as a cancer biomarker [221,222,223], demonstrating the potentially underappreciated mechanisms of TEs in cancers.

## 5. Therapeutic Potential of the LncRNA–Transcription Factor Axis in Cancer Treatment

Although the concept of personalized patient therapy is not new, precision oncology only recently became feasible through the development of genomics approaches that molecularly profile tumors and identify targetable alterations. Thus far, much emphasis has been placed on proteins in cancer therapeutics. However, proteins are not necessarily ideal therapeutic targets, as they often have ubiquitous and essential functions in tissues. LncRNAs could be attractive diagnostic or prognostic biomarkers given their expression specificity, functional diversity, easy detection in bodily fluids, and their deep involvement in cancer regulatory networks. Their regulatory relationship with TFs occur for ~20% of known oncogenes, making this an especially promising vulnerability for therapeutics. Thus far, however, while many ongoing clinical trials are exploring the value of specific lncRNAs as biomarkers, such as CCAT1 in colorectal cancer [224], only prostate-specific PCA3, whose high levels in prostate cancer can be detected in urine, has been clinically approved for diagnostic use [225].

LncRNAs are also being considered as potential therapeutic targets, with a few methodologies having been developed for such treatment: antisense oligos (ASOs), nanoparticle-delivered siRNAs, and small-molecule inhibitors. ASOs and siRNAs bind target lncRNA sequences to promote degradation, making them useful in cancer contexts in which their aberrant upregulation promotes pathology. Small molecule inhibitors, however, target lncRNA’s secondary and tertiary structures, and they have the potential to impinge on their interactions with other macromolecules, such as interfering with lncRNA–TF complexes. Many pre-clinical studies on cancer treatment through lncRNAs are ongoing. For example, MALAT1 is aberrantly expressed in many cancers, and the benefits of targeting it through ASOs and small molecule inhibitors are being investigated [226,227,228,229]. While such lncRNA-targeting cancer treatment has yet to be approved for clinical use, RNA-targeting therapeutics are being used to target mRNAs for tumor treatment. siG12D-LODER and Apatorsen (OGX-427) target mRNAs encoding mutated KRAS and HSP27, respectively, and are both in Phase II clinical trials [230,231]. The ASO Danvatirsen (IONIS-STAT3-2.5Rx, AZD9150) is also undergoing Phase II trials to treat NSCLC by targeting STAT3 mRNA [232]. RNA-based therapies have not been without challenges, as numerous other clinical trials were halted due to issues with specificity, tolerability, and delivery [233]; for example, the siRNA drug DCR-MYC that targeted MYC mRNA to attempt to treat advanced solid tumors, multiple myeloma, and lymphoma [234]. An alternative treatment mechanism could be targeting lncRNAs that regulate the expression of these mRNAs. By targeting CCAT1-L and MYMLAR, or their interactions with CTCF and PCBP2, respectively, one could potentially influence the expression of MYC specifically in tumor tissues. Nonetheless, several companies such as NextRNA Therapeutics are working to harness this untapped potential for cancer treatment [235].

## 6. Conclusions and Perspectives

LncRNA involvement in virtually all aspects and levels of gene expression regulation is pervasive and undeniable—moreover in cancer. However, the impact of their mis/expression in different contexts, whether it significantly disrupts cell order, and if it can be harnessed for clinical use remains largely unknown. Answers to these questions will require first identifying the full repertoire of lncRNA genes, followed by substantially more individual characterization of their function, the regulatory networks they infiltrate, and the molecular mechanisms underlying their control. It will also require a more fully realized understanding of their potential crosstalk: not merely between lncRNAs and interactors but also between the different functions they might exert across the cell to synergize regulatory responses. Such responses may proceed from the transcriptional output of sole lncRNA genes exhibiting different subcellular localizations. For example, HOTAIRM1 can be stably expressed in its unspliced (nuclear) and spliced (cytoplasmic) forms where they appear to regulate transcription and miRNA sponging, respectively [102,236]. Another example is the SAMMSON lncRNA, found in both mitochondria and nucleoli, where it triggers rRNA transcription in each compartment [237]. In each and any case, lncRNA’s involvement in different locales may help sense and liaise responses across the cell. Whether mediated by one or more different transcripts, lncRNAs might integrate the regulation of various processes controlling cell fate, including DNA imprinting, X inactivation, DNA damage repair and DNA replication [238,239,240,241]. The potential for an extensive lncRNA regulatory network that coordinates and fine-tunes cellular activities in normal and cancer contexts holds much therapeutic possibility. Although it remains yet to be fully appreciated, a picture that predicts future therapeutic applicability is slowly emerging.

## Figures and Tables

**Figure 1 cancers-15-03433-f001:**
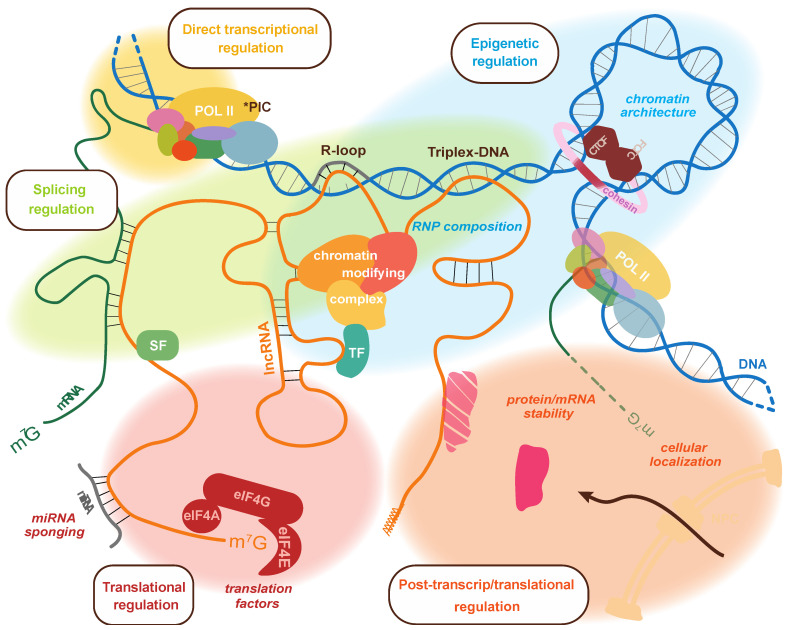
Summary of the processes targeted by lncRNAs to regulate the expression of genes. The different means of control are shown on the same lncRNA molecular to highlight how a single transcript may be capable of multiple regulatory functions. At the level of transcriptional control, they can interact with DNA through the formation of R-loop or triplex-DNA, and they can interact with RNA polymerase II. They can also modulate RNP composition to influence the association of chromatin-modifying complexes as well as 3D chromatin architecture through interaction with CTCF and transcription factors. LncRNAs can also bind splicing factors and mRNAs to regulate splicing. Post-transcriptionally, lncRNAs can regulate mRNA translation by sponging miRNAs, interacting with components of the translation machinery, and even affect the stability or cellular localization of proteins and/or RNAs. * PIC: RNA polymerase II preinitiation complex; TF: transcription factor; SF: splicing factor; NPC: nuclear pore complex.

**Figure 2 cancers-15-03433-f002:**
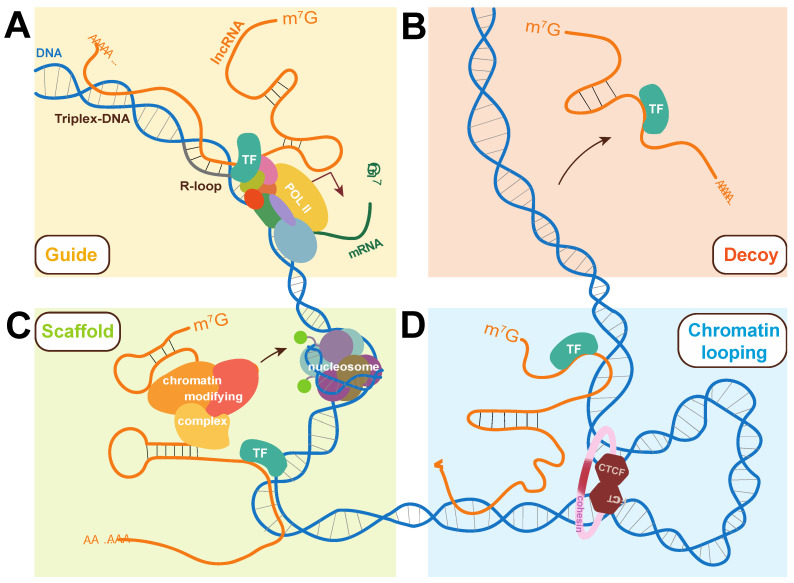
Molecular strategies used by lncRNAs to participate in gene expression regulation. (**A**) LncRNAs can interact with DNA via the formation of triplex-DNA or R-loops to act as guides for transcription factors, influencing their association with chromatin and subsequently transcription. (**B**) LncRNAs have been found to prevent transcription by acting as decoys for transcription factors, binding them to prevent their activation of target genes. (**C**) LncRNAs can simultaneously bind chromatin-modifying complexes and transcription factors, acting as scaffolds in ribonucleoprotein complexes (RNPs). (**D**) Interactions between lncRNAs and nuclear-localized proteins can alter chromatin looping and overall chromatin architecture at virtually all hierarchical levels of 3D chromatin organization. The strategies are shown here in the context of transcriptional control with a focus on their link to transcription factors but roles can be extended to other biological processes. TF: transcription factor.

**Table 1 cancers-15-03433-t001:** A selection of lncRNAs with known roles in cancer.

LncRNA Name	Mode of Action	Associated Pathway	Cancer Context	Ref.
H19	Translational control	Various	Enhances EMT and metastasis in various cancers	[23,24]
PCA3	Translational control	Androgen Receptor (AR) signaling	Prostate cancer biomarker	[25]
HOTAIR	Control of chromatin-modifying complexes	Estrogen Receptor (ER) signaling	Increase invasiveness and metastasis, contribute to chemoresistance in breast cancer	[27,28]
XIST	Spatial chromatin organization	XCI, miRNA regulation	Oncogene in multiple cancers	[29]
GAS5	Transcriptional control	Growth Receptor (GR)	Tumor suppressor	[30,31]
TARID	Control of chromatin-modifying complexes	GADD45A-mediated DNA demethylation	Activation of tumor suppressor TCF21	[32,33]
Khps1	Control of chromatin-modifying complexes	Cell cycle regulation, apoptosis, cell proliferation	Activation of proto-oncogene SPHK1	[34]
MEG3	Control of chromatin-modifying complexes	TGF-β pathway	Associated with repressive chromatin in breast cancer cells	[35]
lnc-LBCS	Control of chromatin-modifying complexes	Epigenetic silencing of SOX2	Inhibit self-renewal and chemoresistance in bladder cancer stem cells	[36]
NEAT-1	Spatial chromatin organization	Transcription regulation, paraspeckle RNA retention	Overexpressed in human tumors and correlated with worse survival	[37,38,39,40]
MALAT-1	Spatial chromatin organization	Transcription regulation, EMT, apoptosis, autophagy	Aberrantly expressed in human cancers, as an oncogene (lung) or tumor repressor (glioma)	[41,42,43,44,45,46,47,48,49]
MALAT-1	Alternative splicing	Influence of splicing factor RSF1, SFPQ activity	Proto-oncogene in hepatocellular carcinoma, promotes tumor growth/metastasis in colorectal cancer	[50,51]
NEAT-1	Alternative splicing	Influence of splicing factor SFPQ activity	Overexpressed in human tumors and correlated with worse survival	[52,53,54]
PNCTR	Alternative splicing	Pro-apoptotic splicing	Overexpressed in variety of cancer cells	[55]
SAF	Alternative splicing	Pro-apoptotic signaling	Apoptotic resistance in human cancer cells	[56]
NAT	Alternative splicing	Snail1-induced EMT	EMT in breast cancer cell lines	[57]
TUG1	miRNA sponging	VEGF-A axis	Increase angiogenesis in glioblastoma, contribute to hypervascularity in hepatoblastoma	[58,59]
TUG1	miRNA sponging	SOX2 and MYC expression	Upregulate abundance of stemness-associated TFs in glioma	[60]
HOXA-AS2	miRNA sponging	EGFR	Vasculogenic mimicry in glioma	[61]
GAS5	Translational control	c-Myc translation	Regulation of c-Myc in lymphoma cell lines	[62]
treRNA	Translational control	E-cadherin translation	Upregulated in breast cancer primary and lymph-node metastasis	[63]
NORAD	Protein stability	DNA damage, genomic stability	Often dysregulated in cancers	[64,65]
SNHG15	Protein stability	Slug signaling	Promotes colon cancer proliferation	[66]
PSTAR	Protein stability	hnRNPK, p53 interaction and cell-cycle arrest	Tumor suppressor in hepatocelluilar carcinoma	[67]

Note: Row color corresponds to “Mode of Action” as depicted in Figure 1.

**Table 2 cancers-15-03433-t002:** A selection of lncRNAs with known transcription factor binding partners.

LncRNA Name	Molecular Function	Transcription Factor Binding Partner	Target Site of Regulation	Effect on Expression	Ref.
GAS5	Decoy	GR	GR targets	Increase or decrease GR-target genes	[31]
TARID	Guide	GADD45A	TCF21	Activation through DNA demethylation	[33]
linc-YY1	Decoy	YY1	Target promoters	Eviction of YY1-PRC2 complexes	[85]
TUG1	Scaffold	YY1	YY1 target genes	Recruitment of PRC2 to silence expression	[60]
PANDA	Decoy	NF-YA	p53 target genes	Prevention of p53-mediated activation of target genes	[177]
	Scaffold	SAFA	senescence-promoting genes	Recruitment of PRCs to repress target genes	[179]
KPRT4	Guide	YB-1	KL5F	Recruitment to 5’ end to enhance transcription	[180]
HAND2-AS1	Unknown	E2F4	C16orf74 promoter	Downregulation	[181]
HNF1A-AS1	Unknown	PBX3	OTX1	Upregulation	[182]
MAGI-AS3	Decoy	HEY1	ACY1 promoter	Increased	[183]
TP53TG1	Decoy	YBX1	PI3K	Decreased	[184]
lncRNA-HIT	Guide	ZEB1	CDH1 promoter	Unknown	[185]
SATB2-AS1	Guide	GADD45A	SATB2	Gene activation through recruitment of WDR5	[186]
LNMAT2	Guide	hnRNPA2, B1	PROX1 promoter	Transcriptional activation	[187]
CISAL	Guide (?)	GABPA	BRCA1 promoter	Sequester TF away to prevent transcriptional activation	[188]
DINO	Scaffold	p53	p53 targets	Increased expression of p53 target genes	[189]
ANRIL	Scaffold	CBX7	INK4a	Recruitment of PRC1 to silence expression	[190]
	Scaffold	EZH2	KLF2 and p21 promoters	Recruitment of PRC2 to silence expression	[191]
	Scaffold	YY1	IL6 and IL8 promoters	Recruitment of YY1 to increase expression	[192]
CCAT1-L	Modulate chromatin folding	CTCF	MYC locus	Mediates chromatin loop to upregulate MYC	[193]
MYMLAR	Modulate chromatin folding	PCBP2	MYC locus	Facilitate enhancer-promoter loop to activate MYC	[194]

Note: Row color corresponds to “Molecular Function” as depicted in Figure 2.

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
