# Peer review of "The Talented LncRNAs: Meshing into Transcriptional Regulatory Networks in Cancer"

_cancers, 2023, doi:10.3390/cancers15133433_

Round 1

Reviewer 1 Report

Studies have found that LncRNAs are often aberrantly expressed in various types of cancer and can both act as an oncogene or tumor suppressor. In this manuscript, Dana and Josee reviewed the current research progress related to well-documented lncRNA, and they particularly emphasized its functions in the context of various cancers and the potential usage of lncRNA for therapeutic treatment of cancers.

Overall, this is a very comprehensive review and very informative. It will be a good resource for researchers who are working in the research field of basic molecular biology and cancer biology. The manuscript is well-written, and the figures are exquisite and well-prepared. I suggest the authors to cut down the numbers of reference, and only cited the important references which provided the original findings. In addition, Section 2.3.1 and 2.3.2 seem too long, especially for the introduction of MALAT1and Xist in section 2.3.2, please simplify.

Author Response

Re: Manuscript Cancers-2444208                                

June 22, 2023

Dear Reviewer,

          Thank you for taking the time to consider our manuscript titled “The Talented LncRNAs: How They Mesh into Transcriptional Regulatory Networks in Cancer”. We are pleased that you found “Overall, this is a very comprehensive review and very informative. It will be a good resource for researchers who are working in the research field of basic molecular biology and cancer biology. The manuscript is well-written, and the figures are exquisite and well-prepared.” Found below is a point-by-point response to your comments. We also explain the changes made to the manuscript.

1. I suggest the authors to cut down the numbers of reference, and only cited the important references which provided the original findings.

response: As suggested, we revised our manuscript by removing extraneous or redundant references. We focused more on citing original findings and limit examples and have thus reduced the total number of references from 276 to 241.

2.
“Section 2.3.1 and 2.3.2 seem too long, especially for the introduction of MALAT1and Xist in section 2.3.2, please simplify.”

response: We have thusly revised Sections 2.3.1 and 2.3.2, removing peripheral details to make the text more concise and simplified. We particularly focused these edits on the introduction of MALAT1, NEAT1, and Xist and were able to reduce the length of these sections by 47 lines of text.

          We highlighted in yellow the sections that we modified. We hope that you will find our answers to your comments suitable and the revised manuscript acceptable for publication in Cancers.

Sincerely,

Josée Dostie, Ph.D.

Professor

Department of Biochemistry &

Rosalind and Morris Goodman Cancer Institute

McGill University

Montréal, Québec, Canada

H3G1Y6

Reviewer 2 Report

The manuscript describes role of Long coding RNAs in regulation of gene expression in cancers quite well and in a comprehensive manner. I would like to suggest few changes as follows -

Title should be revised so that its more crisp. Abstract reads more like information section - should be shorter. Figures are good though use of so many colors should be avoided and more emphasis should be on text so that figure is more clear and self-explanatory. Figure 2 should be divided into multiple figures and more details should be included in them so that they support the text. On page 5, role of Lnc-RNA-PcG is described in regulation of gene expression in a plant Arabidopsis thaliana which is not within the scope of this manuscript. it should be removed. Role of LncRNAs in cancers is explained very briefly in this section, it should be more elaborate. Table 1 should be after table 2 and ref in table have to be numbered serially properly. Section 5 should be little more elaborate. 

Minor english editing is suggested. 

Author Response

Re: Manuscript Cancers-2444208                                

June 22, 2023

Dear Reviewer,

            Thank you for taking the time to consider our manuscript titled “The Talented LncRNAs: How They Mesh into Transcriptional Regulatory Networks in Cancer”. We are pleased that you found “The manuscript describes role of Long coding RNAs in regulation of gene expression in cancers quite well and in a comprehensive manner.” Found below is a point-by-point response to your comments. We also explain the changes made to the manuscript.

  1. “Title should be revised so that its more crisp.”

response: We have modified the title from “The Talented LncRNAs: How They Mesh into Transcriptional Regulatory Networks in Cancer” to “The Talented LncRNAs: Meshing into Transcriptional Regulatory Networks in Cancer” to make the working more concise and crisp.

  1. Abstract reads more like information section - should be shorter.”

response: We have revised the abstract, removing two lines of text to make it shorter. We specifically removed the more detailed information about lncRNAs to make the abstract read less like an information section and flow more like a summary of the review.

  1. Figures are good though use of so many colors should be avoided and more emphasis should be on text so that figure is more clear and self-explanatory.”

response: We edited both Figure 1 and Figure 2 to diminish the background colors, as well as boldened and increased the size of the text. This way, we were able to place more emphasis on the text so that it is more easily explanatory to the reader.

  1. Figure 2 should be divided into multiple figures and more details should be included in them so that they support the text.”

response: We divided Figure 2 into four panels to that the reader is able to more easily interpret the information. Each panel corresponds to a subsection within “3.2. Molecular Function of LncRNAs in Transcription Regulation”, with more detail on each described in the revised Figure 2 legend, therefore better supporting the text.

  1. “On page 5, role of Lnc-RNA-PcG is described in regulation of gene expression in a plant Arabidopsis thaliana which is not within the scope of this manuscript. it should be removed. Role of LncRNAs in cancers is explained very briefly in this section, it should be more elaborate.”

response: We have duly removed the reference to the plant lncRNA APOLO as while it is a good example of R-loop formation occurring in trans, its omission helps the review maintain a tighter focus on human cancers. We also emphasized the tumor suppressor role of TARID and proto-oncogene role of khps1 as they form R-loops and triplex-DNA, respectively, in order to elaborate on the relevance of the mechanisms described in section 2.3.1 in cancers.

  1. Table 1 should be after table 2 and ref in table have to be numbered serially properly.”

response: We re-ordered the rows of lncRNAs in both Table 1 and Table 2 so that they are organized serially according to reference number. Table 1 thus still appears before Table 2 so that the proper serial order is maintained, as Table 1 and 2 correspond to the lncRNAs mentioned in Sections 2 and 3, respectively.

  1. “Section 5 should be little more elaborate.”

response: We have expanded upon Section 5 by elaborating on the limitations of current protein-targeting cancer therapeutics, as well as including examples of RNA-based drugs in Phase II clinical trials. We have also provided an example in which cancer treatment with a failed MYC mRNA-targeting drug could benefit from an approach that takes advantage of the MYC transcription factor’s interactions with lncRNAs.

          We highlighted in yellow the sections that we modified. We hope that you will find our answers to your comments suitable and the revised manuscript acceptable for publication in Cancers.

Sincerely,

Josée Dostie, Ph.D.

Professor

Department of Biochemistry &

Rosalind and Morris Goodman Cancer Institute

McGill University

Montréal, Québec, Canada

H3G1Y6

Reviewer 3 Report

This review about long noncoding RNAs is well organized and written in a way that can be understood by a broad audience. 

My suggestions of minor revisions mostly apply to the figures. I would suggest to the authors use more descriptive figure legends to help walk the reader through the figure. Perhaps using A, B, C, and give a short concise summary of what is being depicted.  Both figure 1 and 2 has some issues with graphic text not in the right position (AAAA..., G7M, etc). It is easier to read descriptions in the legend than going back through the text to reflect on everything that is being summarized.

Lines 711-714 needs citations. This section is not very convincing (lncRNAs for therapeutic target), but it is an interesting concept. 

Author Response

Re: Manuscript Cancers-2444208                                

June 22, 2023

Dear Reviewer,

            Thank you for taking the time to consider our manuscript titled “The Talented LncRNAs: How They Mesh into Transcriptional Regulatory Networks in Cancer”. We are pleased that you found “This review about long noncoding RNAs is well organized and written in a way that can be understood by a broad audience.” Found below is a point-by-point response to your comments. We also explain the changes made to the manuscript.

  1. “I would suggest to the authors use more descriptive figure legends to help walk the reader through the figure. Perhaps using A, B, C, and give a short concise summary of what is being depicted. Both figure 1 and 2 has some issues with graphic text not in the right position (AAAA..., G7M, etc). It is easier to read descriptions in the legend than going back through the text to reflect on everything that is being summarized.”

response: We divided Figure 2 into four panels to that the reader is able to more easily interpret the information. Each panel corresponds to a subsection within “3.2. Molecular Function of LncRNAs in Transcription Regulation”, with more detail on each described in the revised Figure 2 legend, therefore better supporting the text. We also expanded the Figure 1 legend to make it more descriptive and informative. Additionally, we fixed the both figures to make sure all graphic text is in proper alignment.

  1. Lines 711-714 needs citations. This section is not very convincing (lncRNAs for therapeutic target), but it is an interesting concept.”

response: We have expanded upon Section 5: “Therapeutic Potential of the LncRNA-Transcription Factor Axis in Cancer Treatment” by elaborating on the limitations of current protein-targeting cancer therapeutics. We included examples and citations of RNA-based drugs in Phase II clinical trials around lines 711-714. We have also provided an example in which cancer treatment with a failed MYC mRNA-targeting drug could benefit from an approach that takes advantage of the MYC transcription factor’s interactions with lncRNAs. We have thus increased evidential support for lncRNAs as potential cancer therapeutic targets to spark research interest in readers.

          We highlighted in yellow the sections that we modified. We hope that you will find our answers to your comments suitable and the revised manuscript acceptable for publication in Cancers.

Sincerely,

Josée Dostie, Ph.D.

Professor

Department of Biochemistry &

Rosalind and Morris Goodman Cancer Institute

McGill University

Montréal, Québec, Canada

H3G1Y6

Reviewer 4 Report

This is a very thorough review discussing the potential regulatory roles of lncRNAs in cancer pathophysiology. I recommend this review for publication. 

Author Response

Re: Manuscript Cancers-2444208                                

June 22, 2023

Dear Reviewer,

            Thank you for taking the time to consider our manuscript titled “The Talented LncRNAs: How They Mesh into Transcriptional Regulatory Networks in Cancer”. We are pleased that you thought “This is a very thorough review discussing the potential regulatory roles of lncRNAs in cancer pathophysiology. I recommend this review for publication.” We have submitted a revised version of the manuscript based on the comments received from all the Reviewers. We hope that you will agree with the modifications and find the revised manuscript acceptable for publication in Cancers.

Sincerely,

Josée Dostie, Ph.D.

Professor

Department of Biochemistry &

Rosalind and Morris Goodman Cancer Institute

McGill University

Montréal, Québec, Canada

H3G1Y6